# Introducing HoNCAML: Holistic No-Code Auto Machine Learning

Luca Piras[1]  Joan Albert Erráez Castelltort[1]  Cirus Iniesta Carreras[1]  Jordi Casals Grifell[1]
Xavier de Juan Pulido[1]  Marina Rosell Murillo[1]  Cristina Soler Arenys[1]

[1]Eurecat Centre Tecnològic de Catalunya

**Abstract**  In recent years, Machine Learning (ML) has been changing the landscape of many industries, demanding companies to incorporate ML solutions to stay competitive. In response to this imperative, and with the aim of making this technology more accessible, the emergence of "no-code" AutoML assumes critical significance. This paper introduces HoNCAML (*Holistic No-Code Auto Machine Learning*), a new AutoML library designed to provide an extensive and user-friendly resource accommodating individuals with varying degrees of coding proficiency and diverse levels of ML expertise, inclusive of non-programmers. The no-code principles are implemented through a versatile interface offering distinct modalities tailored to the proficiency of the end user. The efficacy of HoNCAML is comprehensively assessed through qualitative comparisons with analogous libraries, as well as quantitative performance benchmarks on several public datasets. The results from our experiments affirm that HoNCAML not only stands as an accessible and versatile tool, but also ensures competitive performance across a spectrum of ML tasks.

## 1 Introduction

In the last few decades, the rise of Machine Learning (ML) applications has revolutionized industry and academia in a wide variety of domains [1]. However, organizations still grapple with challenges in the ML life-cycle [2]. Automated Machine Learning (AutoML) has emerged to address these challenges, offering benefits such as freeing data scientists from tedious tasks and empowering domain experts without requiring them to understand technical complexities. However, democratization of ML is not complete, as most AutoML libraries demand coding proficiency (this is the case, for example, of AutoGluon[3], AutoKeras[4], auto-sklearn[5], Auto-PyTorch[6] and FLAML[7]). This necessity leads to the emergence of the "no-code" and "low-code" paradigms, aiming to make software development more accessible to people with minimal coding skills, particularly in the ML domain [8, 9].

With these principles in mind, we developed an open-source AutoML tool called HoNCAML (*Holistic No-Code Auto Machine Learning*) that aims to provide an extensive and easily-accessible resource to users with varying degrees of coding proficiency and different levels of ML expertise. The term "holistic" stems from the underlying belief that the AutoML problem needs to be addressed as a whole, considering the strict interconnections existing between its sub-problems, rather than treating each task in isolation. This characteristic differentiates HoNCAML from other highly specialized libraries, such as, for example, SMAC [10], which only offers hyper-parameter optimization, NePS[11] and NASLib[12], which focus solely on neural architecture search. Besides, with the term "no-code" we wanted to stress the possibility to execute HoNCAML without writing a single line of code, thus representing a valuable resource for non-programmers or for ML practitioners who seek for quick model prototyping. In this way, HoNCAML lowers the entry barrier for a wide range of users that would be excluded by libraries such as the ones mentioned above.

HoNCAML has been implemented as an open-source Python library and has been released on the Python Package Index (PyPI), from where it can be easily installed. We performed a quantitative

---

evaluation of our tool, consisting in a benchmark on 12 public datasets on classification and regression tasks. Our analysis shows that our library is competitive w.r.t. other open-source AutoML tools, in terms of both performance and efficiency, in many cases surpassing its competitors. In addition to this, we present a qualitative comparison, to highlight the advantages and the limitations of HoNCAML as far as functionalities and user experience are concerned.

The main contributions of this paper are the following: (i) we present HoNCAML, a new no-code AutoML tool that addresses the needs of diverse types of users; (ii) we make the library publicly available on the Python Package Index (PyPI) and release the source code on a GitHub repository, under BSD License[1]; (iii) we provide both a benchmark on an extensive set of public datasets and a qualitative evaluation in comparative terms w.r.t. other AutoML libraries.

## 2 Related work

The numerous surveys and benchmarks published in recent years on AutoML are a proof of the increasing importance of this field. Yao et al. [13] offer a formal definition of AutoML, drawing inspiration from both realms of automation and ML. Zoller et al. [14] start from a rigorous mathematical formulation of the AutoML problem in general and of all the individual tasks that it includes. Karkamer et al. [15] convey an insightful motivation of AutoML by describing in full detail the responsibilities of the data scientist and those of the domain expert, and their interaction within a typical enterprise environment. Barbudo et al. [16] offer one of the most recent surveys, at the time of writing, highlighting the evolution of the field in the last eight years and detecting trends for the future. For more surveys, benchmarks and general perspectives on AutoML, we refer the reader to [17, 18, 19, 20, 21].

Another interesting perspective on AutoML is offered by user studies, that tend to convey a rather qualitative assessment of existing solutions and suggest directions for future research. Sun et al. [22] collected evidence from 19 users revealing that the three major concerns around the AutoML domain are customizability, transparency and privacy, and showing common workarounds to overcome these limitations. In an analogous study, Wang et al. [23] highlight benefits and challenges of AutoML. The authors recommend that AutoML systems should be designed with a mindset of augmenting, rather than automating, the work ML practitioners. Xin et al. [24] carried out a study involving users with varying profiles and degrees of expertise. Their results suggest that AutoML tools should adapt to the proficiency level of the intended user, instead of taking a one-size-fits-all approach. Another important take-home lesson is that an end-to-end solution that handles all stages of the ML workflow in a single environment is the optimal design choice for Auto-ML tools. These theses significantly inspired us in the design of HoNCAML.

Although a great number of specific libraries could be mentioned that play an important role in the AutoML game, we decided to limit the scope of this section to those that fully implement the no-code paradigm. In fact, this characteristic places them as the most direct competitors of HoNCAML. NNI [25, 26] is an open-source toolkit providing functionalities in hyper-parameter optimization, neural architecture search, model compression and feature engineering. H2o [27] is a distributed ML platform including a fully-automated AutoML module. This offers a wide variety of algorithms yielding a healthy amount of diversity across candidate models, which can be exploited by stacked ensembles to produce a powerful final model. TPOT (Tree-Based Pipeline Optimization Tool) [9, 28] represents an original solution w.r.t. the rest of AutoML tools, as it uses genetic programming to optimize ML pipelines [29]. A more detailed comparison of these resources will be presented in Section 4, where they will be compared with HoNCAML under a series of qualitative dimensions. The same Section will also give information about other AutoML libraries that do not follow the no-code paradigm.

---

[1]`https://en.wikipedia.org/wiki/BSD_licenses`

## 3 HoNCAML

HoNCAML is a declarative AutoML framework developed as a Python library with the following fundamental principles in mind: (i) AutoML should be accessible to all kinds of users, from the expert ML practitioners to individuals without any programming skills; (ii) the AutoML problem should be treated in its entirety, taking into account the interdependence between its sub-problems; (iii) to actually foster the democratization of ML, the library should be modular and extensible, and its source code should be open. Several papers in the literature support and provide evidence in favor of the utility of these principles [2, 23, 24].

The tool is designed to accept data in tabular format and provides optimization for a wide range of ML and DL models for regression and classification. It has been developed and tested on UNIX operating system and released as an open-source Python library. The links to the page on the Python Package Index (PyPI), to the GitHub repository and to the full documentations are provided in Appendix A. The rest of this section describes in detail the design, the interface, the tasks and the models provided by HoNCAML.

### 3.1 Design

Under the hood, HoNCAML was designed in a modular and extensible fashion, according to the software design principles of modularity whose benefits have been studied by several works in the literature [30, 31, 32]. In HoNCAML, these principles are realized in that each module is independent of the others and new pieces can be plugged in as desired in a relatively easy way. Besides simplifying the development of the library, this approach is intended to lower the entry barrier for ML engineers who want to customize it or extend its functionalities (e.g., by adding new types of models, new metrics, etc.).

It should be noted that, even in its base version, HoNCAML allows relatively easy integration with other popular ML resources. In fact, the implementation of many of its functionalities is based on open-source libraries scikit-learn[2] and PyTorch[3]. As a consequence, the parameters of the pipelines that can be executed within HoNCAML reflect in most part the interface of the classes provided by these libraries. Furthermore, the models that are trained and returned by HoNCAML are instances of scikit-learn and PyTorch models, which enables an almost seamless integration with those toolkits. These models are saved locally to files and can be loaded through the Python package *pickle* to be used in external programs.

The main components of the library are the following: `Execution`, `Pipeline` and `Steps`. In this hierarchy, `Execution` is at the top level, as it controls the flow of the whole experiment. In fact, this module specifies which pipelines will be run and can include several pipelines at once (for example, run a benchmark and then train the best model with all the data). Lower in the hierarchy, the `Pipeline` module defines each single pipeline, which can be either *train* (train a model with specific parameters), *predict* (predict values using a trained model) or *benchmark* (evaluate several models and parameters to find the one with the best performance). Finally, each pipeline can be composed of one or more `Steps`, which can be of three different types: `DataStep` specifies parameters for data ingestion, pre-processing and storage; `ModelStep` regards model configuration, optimization and handling; finally, `BenchmarkStep` includes settings for benchmark management. Figure 1 illustrates a conceptual diagram that resumes the typical workflow of the tool.

### 3.2 Interface

Inspired by the principles of the no-code paradigm, HoNCAML's interface was designed with the idea to address the necessities of diverse types of users, from the non-programmers to the ML practitioners, and to represent a valid option for quick model prototyping. Three different

---

[2]https://scikit-learn.org/
[3]https://pytorch.org/

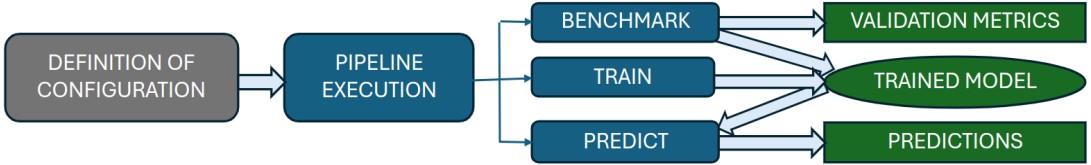

Figure 1: Conceptual diagram illustrating the typical workflow of HoNCAML, from the definition of the configuration up to the output of the pipeline.

modalities of user interaction are available: command-line interface (CLI), graphical user interface (GUI) and source code.

**GUI**. This is the more visually-appealing of the three modalities. This usage is intended for users with limited coding skills or that simply prefer a more graphical approach. It consists of a web-based interface implemented with Streamlit[4], in which the user can upload a dataset, choose the type of task (classification or regression), set up the configuration of the ML pipeline, run the models and evaluate their performance, all in an interactive manner. Different options and configurations are displayed depending on whether the users decide to run a pipeline of type *train*, *predict* or *benchmark*. For users without expertise in ML, a basic default configuration is provided. Configurations are saved to facilitate the reproducibility of the experiments. Appendix A illustrates several captures of the GUI and provides instructions on how to launch it, together with an example of execution.

**CLI**. This interface mode, together with the GUI, is what puts the "no-code" principles into practice in HoNCAML. Compared to the GUI, however, CLI is a more suitable option for slightly more advanced users, who are familiar with the UNIX shell. This modality provides a higher degree of control and customization over the pipeline. A configuration file in YAML format allows the user to choose the type of task, indicate the path to the input data, select the features, define data pre-processing operations such as feature normalization and one-hot encoding, set the hyper-parameter search for a series of ML and DL models with a customized search space, choose a loss function, personalize the optimizer and the evaluation metric, among other things. Trained models are saved in files for later use; reproducibility is ensured by design thanks to configuration files (which store also information about the random seed). In the Appendix A, the reader will find a step-by-step guide on how to get started with HoNCAML through CLI and two examples of configuration files.

**Source code**. Developers and ML practitioners who desire to extend HoNCAML's functionalities or integrate the library into their own applications have open access to the source code of the library in the GitHub repository (link in Appendix A). Once HoNCAML is installed, functionalities can be imported like normal Python modules. However, at the time of writing, a thoroughly documented API for developers has not been published yet, as we prioritized the former two modalities of interface which enable the no-code paradigm.

### 3.3 Tasks

In terms of standard ML pipeline, HoNCAML is able to solve three fundamental tasks:

- **Hyper-parameter Optimization (HPO)**, which refers to the search of the optimal configuration of hyper-parameters for a given model, defined as the one that leads to the best generalization results on unseen data [33].

- **Algorithm selection**, which, closely related to the former one, consists in choosing the algorithm, among a list of candidates (each trained with its own optimized configuration), that leads to the best performance [34]. This can be considered a meta-learning problem according to [35].

---

[4]https://streamlit.io/

- **Neural architecture search (NAS)**, which can be considered as a special case of the former two tasks, where the configuration to be optimized is the architecture of a neural network [36].

These three problems are strictly interrelated. In fact, the selection of an algorithm over another depends heavily on the selection of hyper-parameters for the two algorithms. This is true for both classic ML and DL models. As a consequence, in accordance with the holistic approach, HoNCAML addresses these three tasks as a more general, unique problem, allowing the user to include both ML and DL models into a single benchmark pipeline. Like many existing AutoML solutions, also HoNCAML addresses algorithm selection and HPO in a joint approach, usually referred to as CASH (Combined Algorithm and Hyper-parameter Selection) [37, 38]. The library offers a wide range of state-of-the-art methods to solve this challenge, including, but not limited to, Tree-structured Parzen Estimator (TPE) [39], CMA-ES [40] and Bayesian Optimization [41]. TPE is the default option, although the user can switch to a different algorithm by editing a configuration file (as showed in Appendix A). In addition, there are multiple available stopper and pruner methods, that can be activated to control the behavior of the optimization methods. Options include Hyperband [42] (which is the default option), BOHB [43] and Population-Based Training [44]. Like for the search algorithm, it is the user's responsibility to pick the preferred optimizer in the configuration, in case they wish to change the default setting. All search algorithms and optimizers are imported from Ray Tune[5], a Python library focused on model optimization.

From a general standpoint, the NAS problem can be decomposed in three parts [45]: (i) search space, consisting in the search of the model architecture; (ii) search strategy, which consists in the way in which the search space is explored; (iii) performance estimation strategy, addressing how to evaluate the performance of candidate architectures. In HoNCAML, the search space of candidate neural network architectures is generated through an abstraction called *block*, which can be defined by the user. Each block is a predefined combination of layers, which can be sequentially structured to form deeper networks. For instance, a block can be composed of a dense layer stacked with a drop-out layer. Each layer within a block may have a series of hyper-parameters to optimize, as well (e.g., number of neurons, drop-out rate, etc., depending on the type of layer). Since the NAS problem is treated within CASH, the search strategies reflect the ones mentioned above (Tree-structured Parzen Estimator, CMA-ES and Bayesian Optimization) and are defined *a priori* by the user (or set to the default option) in the configuration file. This feature is what enables HoNCAML to leverage both ML and DL models in an homogeneous way within the same execution. An additional internal logic has been developed for NAS, which guarantees that useless combinations of layers are discarded (e.g., a series of drop-out layers stacked together). The performance estimation strategy follows the same logic as the one used for CASH. Cross-validation parameters, including number of folds and metrics to compute, are defined through the configuration file. The candidate architecture and weights obtaining the best results are selected. It should be noted that, since HoNCAML, at the time of writing, only focuses on tabular data, the NAS problem at hand is, in fact, a simplified one. For this reason, more advanced strategies, such as DARTS [46], have not been integrated in the current solutions. However, they will be taken into account in the future developments, when textual and multimedia data are included in the pipeline. The implementation of the NAS module is based on the DL library PyTorch.

## 3.4 Models

A wide range of ML models is available in HoNCAML for classification and regression, including, but not limited to, Random Forests [47], AdaBoost [48], Gradient Boosting [49], Support Vector Machines [50], etc. The implementation of these models relies on the Python library scikit-learn. As far as DL is concerned, different types of layers can be combined to form neural architectures.

---

[5]https://docs.ray.io/

(a) List of selected libraries, ranked according to their number of GitHub stars.

| Framework | GitHub link | Papers |
|---|---|---|
| NNI | /microsoft/nni/ | [25, 26] |
| TPOT | /epistasislab/tpot | [28] |
| Auto-Sklearn | /automl/auto-sklearn | [55, 5] |
| Autogluon | /autogluon/autogluon | [3] |
| H2O | /h2oai/h2o-3 | [27] |
| FLAML | /microsoft/FLAML | [7] |
| Auto-PyTorch | /automl/Auto-PyTorch | [6] |
| LightAutoML | /sb-ai-lab/LightAutoML | [56] |

(b) List of selected datasets, split by task type.

| Dataset | Type |
|---|---|
| Abalone | Regression |
| Diamonds | Regression |
| House sales | Regression |
| Moneyball | Regression |
| Online news popularity | Regression |
| Sensory | Regression |
| Ada | Classification |
| Adult | Classification |
| Australian credit | Classification |
| Diabetes | Classification |
| German credit risk | Classification |
| Ozone level 8hr | Classification |

Table 1: Libraries and datasets included in the comparative study.

Available options include dense layers [51], recurrent layers [52], LSTM [53], dropouts [54] and others. The complete list reflects the layers provided by PyTorch[6].

## 4 Evaluation

The objective of this section is twofold: on one hand, we want to assess HoNCAML's characteristics, functionalities and user experience from a qualitative standpoint, by comparing it with other open-source AutoML libraires; on the other hand, since a positive qualitative assessment is of little use without competitive performance, we start by evaluating HoNCAML in a quantitative way, by benchmarking it against the same set of competitors on a series of classification and regression tasks. First and foremost, we will describe the methodology and setup employed in our experiments. Then, we will proceed to illustrate the results of both types of evaluations.

### 4.1 Methodology and experimental setup

In order to select the initial set of candidate libraries to be compared against HoNCAML, we revised AutoML surveys and benchmarks (see Section 2). Not to miss recently released tools, we searched Google with the following query: *automl open source frameworks and libraries*. To guarantee accessibility and reproducibility, we filtered the list of candidates by retaining only libraries that (i) are available as actively maintained open-source software and (ii) have a scientific paper associated to it. The list of AutoML frameworks that, at the time of writing and to the best of our knowledge, comply with the stated requisites, is illustrated in Table 1a.[7]

We compared HoNCAML with all selected frameworks, both in the quantitative benchmark and in the qualitative evaluation. The former is based on a set of classification and regression datasets, which have been selected for being publicly available and for reflecting tasks which are coherent with HoNCAML's objectives. The list of datasets, publicly available on OpenML platform[8], is presented in Table 1b. All datasets can be easily found by inserting their names in the platform's dataset search[9]; each page offers a description of the data and additional meta-information. None

---

[6] https://pytorch.org/docs/stable/nn.html. Although the library already supports the integration of all these PyTorch layers, some of them, like Recurrent Layers, require a previous preparation of the data into sequences, which is not implemented yet at the time of writing.

[7] We invite the reader to note that, since AutoML is a rapidly growing field, it is possible that some new frameworks matching our selection criteria might have appeared or become popular since the publication of the present work. Moreover, many of these frameworks are currently under development, therefore some new features might have appeared meanwhile.

[8] https://www.openml.org/

[9] https://www.openml.org/search?type=data&sort=runs

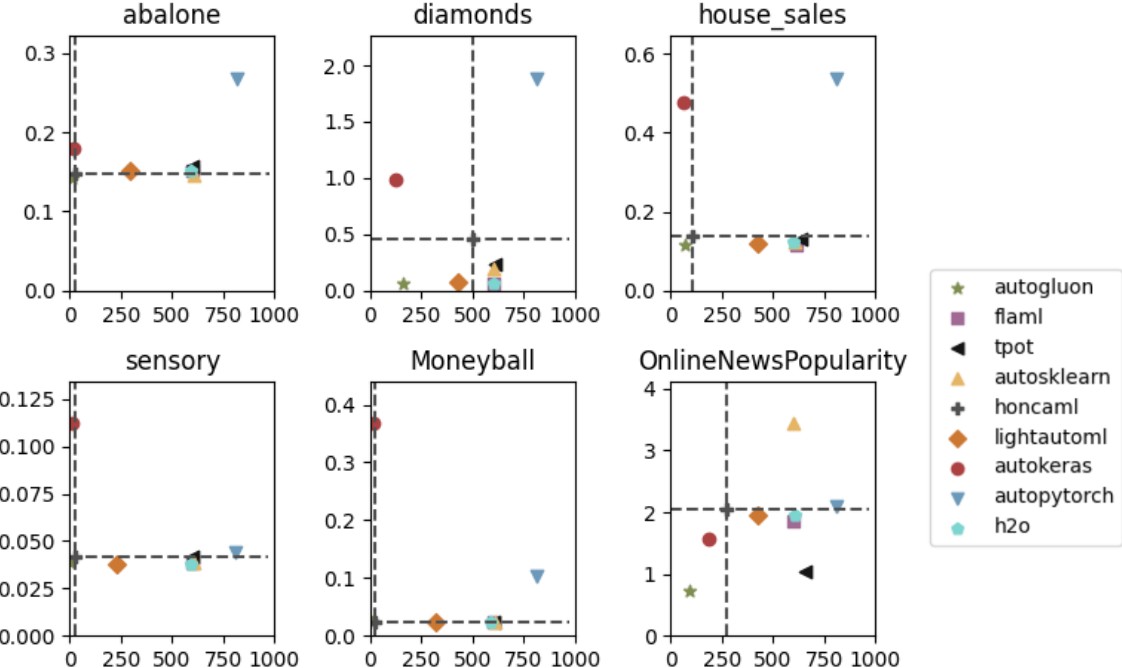

Figure 2: Results of the regression task on the respective datasets. The x-axis of each plot shows the average time, in seconds, that each library took to execute; the y-axis reports the MAPE score. HoNCAML performance is highlighted by the dotted lines.

of the datasets has a license that prevents its usage in a research context like the present study, nor contains personally identifiable information or offensive content. For each dataset, we used a 3-fold cross-validation approach, using the same train and test splits for all libraries, and repeating the execution on each fold with 5 different random seeds, to guarantee robustness of results[10]. Regression results are reported via Mean Absolute Percentage Error (MAPE), while classification has been assessed with macro F1-score, two standard and well-established metrics in their respective domains [57, 58]. Both metrics are expressed as the mean obtained across all folds and random seeds. In addition, we measure the average time that each library takes to complete the execution on each dataset, within a maximum pre-defined time-budget of 600 seconds (inclusive of training and prediction time)[11]. All frameworks have been tested with their default settings, as the idea behind this analysis is to provide a direct, out-of-the-box performance estimation, without any prior settings modifications. Experiments were run on a machine with a 16-core Intel Xeon Processor and 32GB of RAM.

## 4.2 Quantitative benchmark

In interpreting the results of the quantitative benchmark, the reader should bear in mind that our objective was not to develop the best performing AutoML library, but to offer an accessible and versatile tool that guarantees competitive results compared to other available solutions. The code for running the experiment, together with requirements, instructions, and raw results, is available on HoNCAML's repository: `https://github.com/Data-Science-Eurecat/HoNCAML/tree/feature/paper/paper`.

Results on both regression and classification tasks indicate that, overall, HoNCAML stands in a competitive position w.r.t. the other libraries, both in terms of accuracy and efficiency. In Figure 2, illustrating the regression results for each respective dataset, HoNCAML tends to place itself

---

[10]Due to internal technical complexities, it was not possible to execute NNI in a cross-validation setting. For this reasons, results for this library will not be reported in the quantitative benchmark

[11]Some executions of auto-PyTorch and TPOT, due to their internal functioning, exceed the time limit.

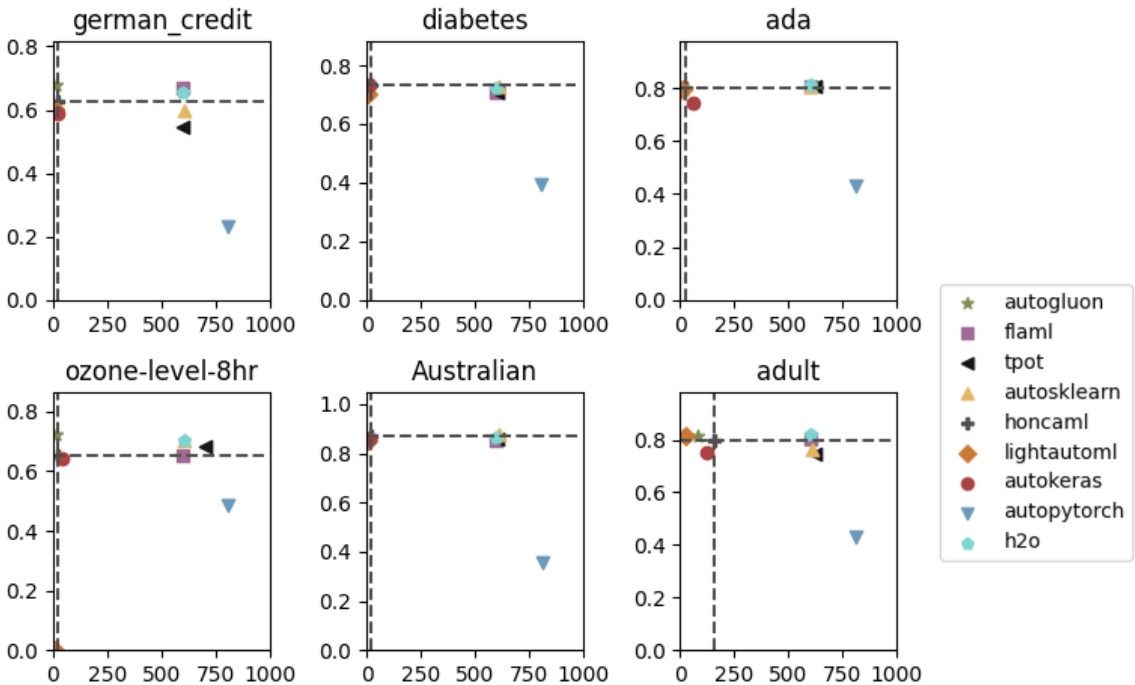

Figure 3: Results of the classification task on the respective datasets. The x-axis of each plot shows the average time, in seconds, that each library took to execute; the y-axis reports the F1-score. HoNCAML performance is highlighted by the dotted lines.

around the lower-left corner of the plots, which means that it is able to obtain low MAPE scores in a relatively short time (low MAPE indicates good results). This is especially true for *Abalone*, *House sales*, *Sensory*, and *Moneyball*; on these datasets, in fact, MAPE scores are aligned with the best ones and are obtained in a significantly shorter time. Even on *Diamonds* and *Online news popularity*, which are the datasets on which HoNCAML struggles the most, its performance is close to the competitors' average both for speed and accuracy. This finding highlights the solidness of our solution. Classification results put HoNCAML in an even better light (Figure 3). Our library lies in the upper-left corner of the plot for all datasets, as both its F1-scores and running times are among the best ones in all cases (high F1-scores indicate good results). The direct comparison with the other no-code tools, namely TPOT and H2o, indicates for HoNCAML a similar performance in terms of MAPE and F1-score. However, when it comes to speed, our library clearly outperforms the other two libraries on all 12 datasets, while achieving metrics in a comparable range.

Another positive finding is the relatively low variance observed across folds and random seeds, which suggests the stability of the training and optimization components (Tables 2 and 3 of Appendix B). We tested statistical significance of these results through the Friedman's test, which is a non-parametric statistical test used to determine whether there are statistically significant differences between the means of three or more paired groups [59]. In our case, each set of MAPE and F1-scores, respectively, obtained by a library corresponds to a group. The two tests on regression and classification reported p-values of 0.002 and 0.001, respectively. Therefore, we can claim with sufficient confidence that the difference in performance obtained by each library across datasets is statistically significant.

## 4.3 Qualitative comparison

Now that we have demonstrated that HoNCAML's performance is competitive w.r.t. other well-established AutoML libraries, in this section we set out to present a qualitative comparison. HoNCAML and the other libraries will be analyzed under a series of dimensions relative to user

experience and offered functionalities. The comparison is supported by supplementary tables attached in Appendix B.

**4.3.1 Interface**. The interface dimension marks a split between frameworks that focus on interactive and visual interfaces and others that prioritize the access through developers' API. As stated above, CLI and GUI are the two ways that empower the no-code paradigm. Only HoNCAML, NNI, TPOT, and H2o implement at least one of these two interface types (Table 4, in Appendix B). HoNCAML is the only library that offers both, thus standing out as the most accessible and versatile resource in the no-code category[12]. The most evident limitation of HoNCAML, under the interface dimension, is its lack of a developers' API. This feature, which is a work in progress at the time of writing, has been under-prioritized so far to the advantage of functionalities that allow the no-code paradigm. The most complete framework, in terms of API variety, is H2o, which provides its services through a Python, R and Java API and, in addition, as a REST service. All others libraries, except Auto-PyTorch, provide a Python API.

**4.3.2 Models**. In studying the models implemented by the frameworks under analysis, we decided to create a distinction between DL and ML models. In spite of the remarkable success of DL in many areas and tasks [60], traditional ML still represents a valid option in many scenarios, for example, when the data is structured (tabular) and/or limited [61]. Furthermore, ML has the advantage that, normally, it is significantly less time- and hardware-consuming compared to DL. For these reasons, we believe it is important not to discard older ML models even in modern AutoML tools. As it can be observed in the two right-most columns of Table 4 (Appendix B), HoNCAML is one of the most complete libraries, as it covers both areas. Almost all competitors include ML models in their workflow, with the exception of AutoKeras and Auto-PyTorch, which are solely focused on DL. On the other hand, only Auto-Sklearn does not provide algorithms in the DL domain, if we exclude the Multi-layer Perceptron class, belonging to the scikit-learn library.

**4.3.3 Steps**. Hereby we study which steps of the typical ML pipelines are covered by each library, considering pre-processing, feature engineering, model selection, HPO (these latter two often addressed jointly as CASH) and NAS (Table 5 in Appendix B). It can be claimed that the inclusion of several ML steps within the same library strongly affects usability and accessibility. In fact, this represents an advantage for basic users who are interested in a black-box usage, as they do not have full conscience of all the steps involved in the typical ML pipeline. At the same time, expert users may benefit from a unified resource reducing the overhead generated by switching between different tools specialized on specific tasks (such as the aforementioned SMAC, NePS or NASLib). Several works in the literature further support the benefits of this holistic approach [24, 19].

Pre-processing and feature engineering are two important parts of any ML-related development, yet they are covered only by approximately half of the frameworks [13]. On the other hand, it is notable that all frameworks offer model selection and HPO techniques, which comes to highlight that these two components of the pipeline enjoy a paramount importance within the AutoML world. Not surprisingly, all frameworks that implement DL models (see Table 4 in Appendix B) also provide NAS functionalities, as these are crucial to optimize the hyper-parameter space and the architecture of DL models. Also in this aspect, HoNCAML is well-positioned w.r.t. its competitors, as it offers functionalities in all the steps, except for feature engineering, which will be addressed in the future.

---

[12]Although the general framework of H2o can be executed via command line (as illustrated here `https://github.com/h2oai/h2o-3/blob/master/h2o-docs/src/product/howto/H2O-DevCmdLine.md`), this is not true, at the time of writing, for its AutoML module (`https://docs.h2o.ai/h2o/latest-stable/h2o-docs/automl.html`.

[13]It should be mentioned, however, that DL-only frameworks have less need of this part of the pipeline, insofar it is implicitly done through the neural networks themselves [62]. This is the case of AutoKeras and Auto-PyTorch

## 5 Broader Impact Statement

The development and deployment of the proposed AutoML library carry both promising benefits and potential considerations across environmental, ethical, and societal dimensions. On the positive side, by democratizing ML model creation and deployment, the tool has the potential to significantly lower the barrier of entry for individuals and organizations to harness the power of Artificial Intelligence (AI). However, the widespread adoption of such tools also raises ethical concerns related to the responsible use of AI technology. As far as privacy is concerned, HoNCAML is completely transparent, since its code is open, it is installed and run on local machines and does not collect any personal data. On the other hand, it is imperative to address issues such as bias in model outcomes and potential misuse of automated decision-making. In Section 6 we mention mitigation strategies that we intend to consider in our future work. Moreover, the environmental footprint of increased AI model training and deployment should be carefully considered, as resource-intensive processes could contribute to a rise in energy consumption and electronic waste.

## 6 Conclusions

In this paper we presented HoNCAML, a new open-source library that addresses the AutoML problem from a holistic perspective through a no-code approach. After presenting the design principles underlying the library and describing its technical and theoretical details, we compared HoNCAML against a set of other open-source libraries, both in qualitative and quantitative terms. Our evaluation shows that HoNCAML is an accessible, versatile and complete tool, at the same time ensuring competitive performance across a spectrum of ML tasks. If we limit our comparison to other no-code libraries, HoNCAML is the only one offering both a CLI and a GUI, thus standing out as the most complete no-code AutoML tool in terms of user interface. In addition to this, the quantitative benchmark highlighted that HoNCAML is more efficient than the other no-code libraries, as it achieves competitive metrics in classification and regression tasks in shorter time.

It should be stressed that versatility has to be intended mostly as far as user types and interfaces are concerned. Indeed, one of the main limitations of HoNCAML is that, at the time of writing, it is unable to handle images, texts or time-series data (unless the user previously converts them into a tabular data structure). In the future, we plan to extend the library so as to provide classification and regression capabilities also for these types of data. Another important step on our future road-map is the implementation of an extensive feature engineering module, which would allow the user to apply common transformations to data, select features based on estimated importance and create custom features. Another functionality that would further improve HoNCAML's versatily and accessibility is a fully-documented developers' API. Finally, we are aware of the importance that trustworthiness, transparency, fairness and explainability have in the development and usage of a tool like HoNCAML [63, 64, 65]. In the future, we plan to integrate algorithms and techniques to detect and mitigate biases in both the training data and the models generated, drawing inspiration from promising research that points into the same direction [66, 67, 68]. This could involve measures such as fairness-aware learning algorithms, bias detection tools, and techniques for de-biasing data and models [69, 70], as well as visualization tools that allow delving deeper into the functioning of models [71, 72].

**Acknowledgements**. This work was financially supported by the Catalan Government through the funding grant ACCIÓ-Eurecat (Project TRAÇA-HoNCAML).

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

## A How to run HoNCAML

In this appendix we will illustrate how to install HoNCAML and how to execute it through command-line interface (CLI) and graphical user interface (GUI).

The source code is publicly available in a GitHub repository: `https://github.com/Data-Science-Eurecat/HoNCAML`

The library has been published on the Python Package Index (PyPI): `https://pypi.org/project/honcaml/`

For an exhaustive explanation of execution modalities and implementation details, we refer the reader to the official documentation: `https://data-science-eurecat.github.io/HoNCAML/`

### Install

To install HoNCAML, it is required to have Python installed (version 3.10) on a UNIX operating system. The installation command is the following:

```
pip install honcaml
```

### GUI

To start the GUI, it is sufficient to run the following command:

```
honcaml --gui
```

This will open a web application on the predefined browser, running on localhost (port 8501 by default). In the upper part of the page (Figure 4), the user can choose whether he wants to introduce configurations manually via the web app, or by pasting or uploading some previously edited YAML file (in this section, we will show functionalities referring to the manual insertion of configurations; next sections will illustrate how to generate and edit YAML configuration files thorugh the CLI). In addition, the user has the option to use basic or advanced configurations, can select the type of pipeline he want to execute, upload the dataset and select the target variable. A preview of the dataset will be shown in a tabular format.

Figure 4: Preliminary options of the GUI.

In the next step (Figure 5), the user is invited to select the type of problem they want solve (regression or classification). A warning will be shown in case the chosen problem is not coherent with the target variable previously indicated (for example, choosing the *price* variable, of type float, with a classification problem would lead to a warning). Moreover, the user can select the subset of variables to include in the training as features and can choose to apply normalization to some of them.

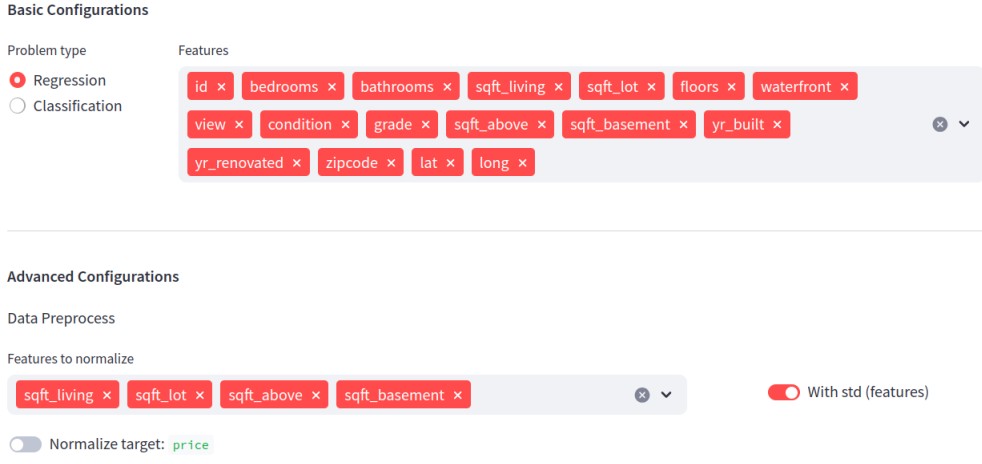

Figure 5: Basic and advanced pipeline configuration.

Following, the user is guided through the selection of models they desire to include in the pipeline (Figure 6. In this example, since we are illustrating the execution of a benchmark, it is possible to include different models at the same time for performance comparison. For each model, the user can adjust the range of values to be searched in the optimization phase.

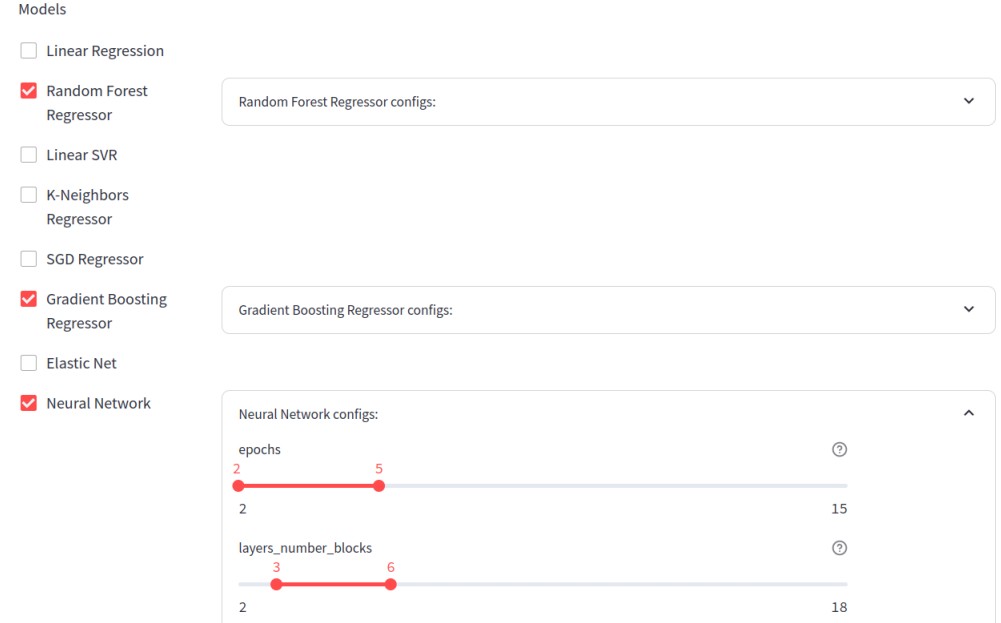

Figure 6: Selection of models and hyper-parameters. This figure has been cut for convenience, but many more options are available for the Neural Network model

Another section of the GUI lets the user pick the metrics he wants to include in the benchmark and the one that should guide the tuner in the optimization process (Figure 7).

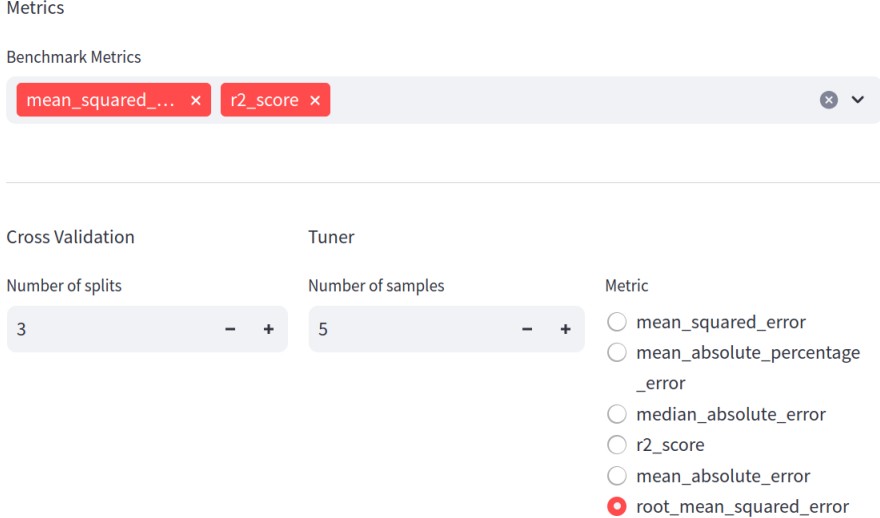

Figure 7: Selection of metrics for optimization and evaluation

After pressing the *Run* button, the execution of the pipeline will start. The process will take more or less time depending on the size of the data, the pipeline type, the selected problem and the number of models included in the analysis, among other factors. At the end, results will be presented in a tabular format or through charts, and, in case of running a benchmark, the best model and set of hyper-parameters will be pointed out (Figure 8). In addition, the user has the chance to download results as a .csv file.

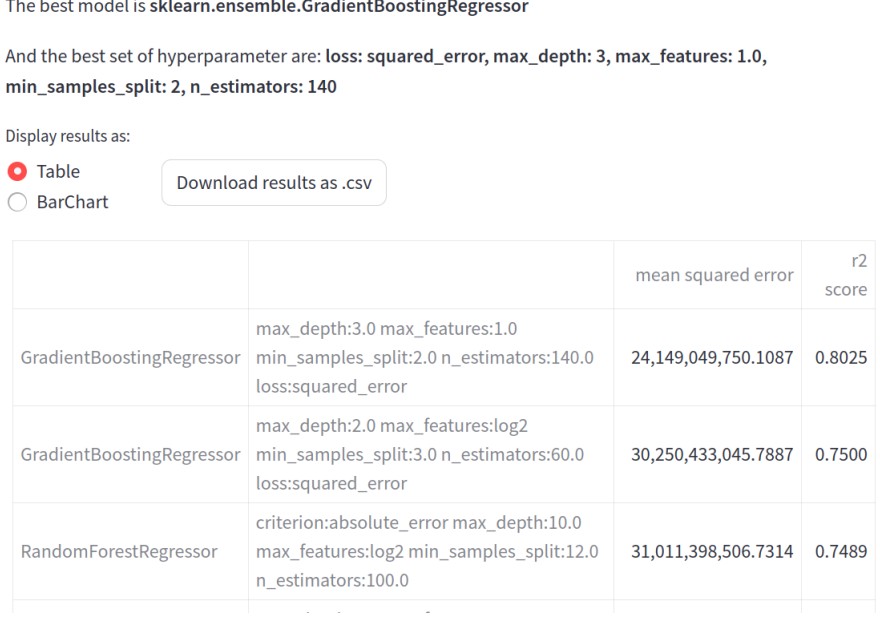

Figure 8: Presentation of results

**CLI usage**

**Executing an example with sample data and configurations.** To get a glimpse of HoNCAML with example data and configuration, it is sufficient to run:

```
honcaml --example <example_directory>
```

Please replace <example_directory> with the directory name that suites you the most. This command creates a directory containing sample data and configuration files, which allows the user to get started with HoNCAML in a straightforward manner. Inside the new folder, the user will find two sub-directories:

- `data`, which contains sample training and test datasets for classification and regression, in `.csv` format;

- `files`, providing sample configuration files for both aforementioned ML tasks, in YAML format.

To run a quick example without delving into the details of these two folders, it is enough to enter the specified directory (`cd example_directory`) and execute one of the configurations located in `files` directory. For example, to execute the benchmark for a classification task, launch the following command:

```
honcaml --config files/classification_benchmark.yaml
```

This instruction will train, optimize and test a series of models on a sample dataset. The CLI will inform the user of the operations that are being performed and of the results. As a consequence, a new directory will be generated with relative path `honcaml_reports/<execution-id>`, where <execution-id> is a code assigned to the current execution, obtained by concatenating date and time of the execution. Inside this folder, `results.csv` will keep trace of the metrics obtained by each model. Furthermore, a file named `best_config_params.yaml` will store the settings of the model that led to the best results, together with its hyperparameters.

**Executing pipelines with custom configurations.** In a realistic scenario, the user typically wants to run HoNCAML on their own dataset, specifying a custom configuration. In order to facilitate the creation of the configuration file, the CLI offers the possibility to generate a template that the user can fill according to its use case. This can be done through two different alternative commands, depending on whether a basic or and advanced configuration file is desired:

- Basic: honcaml –generate-basic-config <config-file-name> –pipeline-type <pipeline-type>

- Advanced: honcaml –generate-advanced-config <config-file-name> –pipeline-type <pipeline-type>

The parameter <config-file-name> is the name (including the path) of the new template that will be generated. Allowed values for <pipeline-type> are `train`, `predict` and `benchmark`. The difference between the output of these two commands lays in the number and kinds of parameters included in the template. For a detailed description of possible configurations, we refer the reader to `https://data-science-eurecat.github.io/HoNCAML/configuration.html`.

Once the YAML template has been generated, the user should edit it with the settings that best suite their objective. As a next step, it is possible to execute the pipeline defined by the new configuration:

```
honcaml --config <config-file-name>
```

After running a pipeline of type `train`, a new model object will be saved in the folder `honcaml_model` (at the root of `example_directory`). At this point, a `predict` pipeline can be run to apply the previously trained model on the data specified in the configuration file. The predictions will be saved in a file named `predictions-<execution-id>.csv`, in the `honcaml_reports` directory.

**Configuration files**

We hereby illustrate some examples of configuration files, to better understand their structure and the types of functionalities they enable. A basic YAML for a `train` pipeline of a classifier looks like the following:

```
global:
  problem_type: classification

steps:
  data:
    transform:
        encoding:
          OHE: True
    extract:
      filepath: data/classification_train.csv
      target: class

  model:
    transform:
      fit:
        estimator:
          module: sklearn.ensemble.RandomForestClassifier
          params:
            n_estimators: 100
        cross_validation:
          module: sklearn.model_selection.KFold
          params:
            n_splits: 2
            shuffle: True
            random_state: 90

    load:
      filepath: honcaml_models/sklearn_classification.sav
```

The `global` section is used to specify generic parameters. In this case, a classification task has been selected. The section `steps` contains information on the modular steps included in the pipeline, from `data` loading and transformation up to `model` optimization (under the respective tags). Under `data/extract` the user can indicate the `filepath` of the training dataset and the column to use as `target` to predict. Under the `data/transform` tag, it is possible to specify data pre-processing operations, like, in this case, one-hot encoding (OHE). As for `model` definition, the basic YAML allows to specify parameters for training and validation under the `transform/fit` tag. In this example, the user has chosen to train a Random Forest classifier from the scikit-learn library with 100 estimators (i.e., trees of the forest). The optimization of the model is performed through cross-validation, relying on the KFold `module` provided by scikit-learn, with the parameters

specified under the params tag. Finally, it is possible to insert the `filepath` in which to save the trained model for later use, in the `model/load` section.

It is also useful to see an example of configuration file for a pipeline of type benchmark, since this allows performing hyper-parameter optimization and model selection among a list of candidates.

```
global:
  problem_type: classification

steps:
  data:
    extract:
      filepath: data/classification_train.csv
      target: class

  benchmark:
    transform:
      models:
        sklearn.ensemble.RandomForestClassifier:
          n_estimators:
            method: randint
            values: [ 2, 110 ]
          max_features:
            method: choice
            values:
              - sqrt
              - log2
              - 1 # It means 'auto'
        sklearn.neighbors.KNeighborsClassifier:
          n_neighbors:
            method: randint
            values: [ 1, 100 ]
          weights:
            method: choice
            values:
              - uniform
              - distance

    cross_validation:
      module: sklearn.model_selection.KFold
      params:
        n_splits: 2
        shuffle: True
        random_state: 90

    tuner:
      search_algorithm:
        module: ray.tune.search.optuna.OptunaSearch
        params:
      tune_config:
        num_samples: 5
```

```
          metric: roc_auc_score
          mode: max
        run_config:
          stop:
            training_iteration: 2
        scheduler:
          module: ray.tune.schedulers.HyperBandScheduler
          params:

    metrics:
      - accuracy_score
      - f1_score:
          average: macro

  load:
    save_best_config_params: True
    path: honcaml_reports

model:
  transform:
    fit:
      cross_validation:
      module: sklearn.model_selection.KFold
      params:
        n_splits: 2
        shuffle: True
        random_state: 90

  load:
    filepath: honcaml_models/sklearn_classification_benchmark.sav
```

In this advanced YAML file, the user has decided to execute a benchmark pipeline to compare a Random Forest and a K-Neighbors models (defined under benchmark/transform/models for a classification problem. For each model and each parameter, the user defines the range of values and the method to use to perform the search. The tuner section specifies a series of settings which are used by the optimization module (based on Ray Tune library), including the search_algorithm, the tune_config, the run_config and the scheduler. Under the metrics tag, the user can indicate the metrics that should be used to guide the optimization. The best model with the best settings will be stored in the filepath specified under model/load.

More examples of configuration files can be found in the GitHub repository: https://github.com/Data-Science-Eurecat/HoNCAML/tree/main/honcaml/config/examples/files. The official documentation provides full details on basic and advanced configuration files for all possible pipelines: https://data-science-eurecat.github.io/HoNCAML/configuration.html.

## B Supplementary tables

Hereby we attach supplementary tables, illustrating the variance of the metrics obtained in the experiments of Section 4.2 (Tables 2 and 3) and the qualitative comparison of Section 4.3 (Tables 4 and 5).

Table 2: Variance from the mean MAPE across folds and random seed, for the regression task. Values smaller than 0.001 are reported as 0.

| | Dataset | | | | | |
|---|---|---|---|---|---|---|
| Library | Abalone | Diamonds | House sales | Moneyball | Online news | Sensory |
| HoNCAML | 0.001 | 0.06 | 0.01 | 0 | 0.06 | 0.004 |
| TPOT | 0.003 | 0.01 | 0.001 | 0 | 0.02 | 0.001 |
| AutoKeras | 0.003 | 0.17 | 0.13 | 0.24 | 0.04 | 0.09 |
| Auto-Sklearn | 0.001 | 0.001 | 0 | 0 | 0.6 | 0 |
| H20 | 0.005 | 0 | 0.001 | 0 | 0.03 | 0.001 |
| Autogluon | 0.001 | 0 | 0 | 0 | 0.03 | 0.001 |
| FLAML | 0 | 0.001 | 0.001 | 0 | 0.09 | 0 |
| Auto-PyTorch | 0.23 | 0.23 | 0.05 | 0.01 | 0.05 | 0.001 |
| LightAutoML | 0.001 | 0.001 | 0 | 0 | 0.02 | 0.001 |

Table 3: Variance from the mean F1-score across folds and random seed, for the classification task.

| | Dataset | | | | | |
|---|---|---|---|---|---|---|
| Library | Ada | Adult | Austr. credit | Diabetes | German credit | Ozone level |
| HoNCAML | 0.002 | 0.004 | 0.007 | 0.01 | 0.02 | 0.03 |
| TPOT | 0.002 | 0.001 | 0.004 | 0.01 | 0.01 | 0.01 |
| AutoKeras | 0.007 | 0.002 | 0.01 | 0.02 | 0.02 | 0.02 |
| Auto-Sklearn | 0.002 | 0.001 | 0.01 | 0.004 | 0.02 | 0.02 |
| H20 | 0.003 | 0.001 | 0.005 | 0.008 | 0.02 | 0.01 |
| Autogluon | 0.005 | 0.001 | 0.005 | 0.01 | 0.01 | 0.02 |
| FLAML | 0.002 | 0.003 | 0.01 | 0.002 | 0.01 | 0.01 |
| Auto-PyTorch | 0.16 | 0.04 | 0.07 | 0.05 | 0.03 | 0.05 |
| LightAutoML | 0.001 | 0.001 | 0.001 | 0.001 | 0.01 | 0.001 |

Table 4: Comparison of the libraries according to their interface functionalities and provided models (ML = Machine Learning, DL = Deep Learning).

| | Interface | | | Models | |
|---|---|---|---|---|---|
| Framework | CLI | GUI | API | ML | DL |
| HoNCAML | ✓ | ✓ | in progress | ✓ | ✓ |
| NNI | ✓ | | ✓ | ✓ | ✓ |
| TPOT | ✓ | | ✓ | ✓ | ✓ |
| AutoKeras | | | ✓ | | ✓ |
| Auto-Sklearn | | | ✓ | ✓ | |
| Autogluon | | | ✓ | ✓ | ✓ |
| H2O | | ✓ | ✓ | ✓ | ✓ |
| FLAML | | | ✓ | ✓ | ✓ |
| Auto-PyTorch | | | | | ✓ |
| LightAutoML | | | ✓ | ✓ | ✓ |

Table 5: Comparison of the libraries according to covered steps of the ML pipeline.

| Framework | Pre-proc. | Feature Eng. | Model Sel. | HPO | NAS |
|---|---|---|---|---|---|
| HoNCAML | ✓ | | ✓ | ✓ | ✓ |
| NNI | | ✓ | ✓ | ✓ | ✓ |
| TPOT | ✓ | ✓ | ✓ | ✓ | |
| AutoKeras | | | ✓ | ✓ | ✓ |
| Auto-Sklearn | ✓ | ✓ | ✓ | ✓ | |
| Autogluon | ✓ | | ✓ | ✓ | ✓ |
| H2O | ✓ | | ✓ | ✓ | ✓ |
| FLAML | | | ✓ | ✓ | |
| Auto-PyTorch | | | ✓ | ✓ | ✓ |
| LightAutoML | ✓ | ✓ | ✓ | ✓ | |

