# OpenReview forum: "Introducing HoNCAML: Holistic No-Code Auto Machine Learning"
_automl.cc/AutoML/2024/ABCD_Track — AutoML 2024 (ABCD Track)_

### Official Review · Reviewer_aeb2 · 2024-03-28

**Potential Impact On The Field Of Automl Rating:** 2
**Technical Quality And Correctness Rating:** 2
**Clarity Rating:** 3

**Summary Of Contributions:**

This paper introduces "HoNCAML" which is an accessible and versatile AutoML tool. The tool is compared against analogous libraries, as well as quantitative performance benchmarks on several public datasets.

**Actions Required To Increase Overall Recommendation:**

- More user centered studies comparing other automl tools
- Clearer motivation
- Evaluations on more benchmarks

**Clarity:**

The presentation in the paper is clear in most parts. However, library is not very well documented currently

**Overall Review:**

Positive:
- The paper is clearly written
- The library is under active maintenance and development

Negatives:
- Check "Technical Quality And Correctness"
- Insufficient docs for the library
- Unclear motivation
- Insufficient evaluations on different HPO and NAS benchmarks

**Potential Impact On The Field Of Automl:**

There are a plethora of AutoML libraries available eg: SMAC(https://github.com/automl/SMAC3), NEPS (https://github.com/automl/neps), NASLib (https://github.com/automl/NASLib) and others that the authors mention in their paper. . The first two being quite general to different problem types. Most of these are very user friendly and have a non-steep learning curve. Given plethora of domain-specific (and user friendly) libraries the motivation for a new library is unclear to me.

**Reproducibility:**

The library is open-sourced so I would rate the reproducibility as high in general.

**Review Confidence:**

4

**Review Rating:**

3

**Review Summary:**

Given a plethora of automl libraries (some of which are quite well documented), I find the motivation for the new library unclear. Furthermore I think the work is not very well motivated (also the qualitative metric of "user friendliness" of an automl tool is not evaluated adequately. Also the evaluation on benchmarks is limited.

**Technical Quality And Correctness:**

The presentation and quality of the paper is good in most parts.

The motivation for a new library is unclear to me and I have several questions (1) What are concrete ways in which this library is more easily accessible to a variety of users compared to other libraries presented in the paper and SMAC(https://github.com/automl/SMAC3), NEPS (https://github.com/automl/neps), NASLib (https://github.com/automl/NASLib)? Could the authors tabulate the advantages/disadvantages of each (2) I don't see the motivation behind having "HPO","AC","NAS" in a single library. There are often very specialised methods for each of these tasks (especially for modern deep learning applications) eg: check [1,2,3] for NAS. I think the potential impact of the work can be largely increased by reducing the scope to for eg: only to DL or ML applications and doing a thorough benchmarking there (3) Qualitative metrics like "accessibility", "user-friendliness" are not easily quantifiable. It would be great to run user studies to compare other libraries to HoNCAML in terms of such qualitative metrics. (4) Since NAS is currently a part of the library I think evaluation on various NAS benchmarks (check [4]) is very important to benchmark the library.

[1] Cai, H., Gan, C., Wang, T., Zhang, Z. and Han, S., 2019. Once-for-all: Train one network and specialize it for efficient deployment. arXiv preprint arXiv:1908.09791.

[2] Chen, M., Peng, H., Fu, J. and Ling, H., 2021. Autoformer: Searching transformers for visual recognition. In Proceedings of the IEEE/CVF international conference on computer vision (pp. 12270-12280).

[3] Wang, H., Wu, Z., Liu, Z., Cai, H., Zhu, L., Gan, C. and Han, S., 2020. Hat: Hardware-aware transformers for efficient natural language processing. arXiv preprint arXiv:2005.14187.

[4] Mehta, Y., White, C., Zela, A., Krishnakumar, A., Zabergja, G., Moradian, S., Safari, M., Yu, K. and Hutter, F., 2022. Nas-bench-suite: Nas evaluation is (now) surprisingly easy. arXiv preprint arXiv:2201.13396.

---

### Official Review · Reviewer_ctFw · 2024-04-02

**Potential Impact On The Field Of Automl Rating:** 3
**Technical Quality And Correctness Rating:** 4
**Clarity:** The work is presented clearly.
**Clarity Rating:** 4

**Summary Of Contributions:**

This paper introduces HoNCAML (Holistic No-Code Auto Machine Learning), a new open-source AutoML library designed to provide an extensive and user-friendly resource for individuals with varying degrees of coding proficiency and ML expertise. HoNCAML aims to democratize ML by offering a versatile interface with different modalities tailored to the user's proficiency, including a no-code command-line interface (CLI) and graphical user interface (GUI). The library takes a holistic approach, addressing the AutoML problem as a whole by considering the interconnections between sub-problems. HoNCAML is evaluated both qualitatively, through comparisons with analogous libraries, and quantitatively, via performance benchmarks on several public datasets. The results affirm that HoNCAML is an accessible and versatile tool that ensures competitive performance across a spectrum of ML tasks. The paper also discusses the potential broader impact of the library and its limitations.

**Actions Required To Increase Overall Recommendation:**

* Extend HoNCAML's capabilities to handle a wider range of data types (e.g., images, text, time series) and provide a roadmap for implementation.
* Integrate a flexible and user-friendly feature engineering module that allows users to easily apply common transformations and create custom features within the no-code paradigm.

**Overall Review:**

Strengths:
* HoNCAML's no-code approach and versatile interface options (CLI and GUI) make AutoML more accessible to users with varying levels of coding proficiency and ML expertise. This democratization of ML through user-friendly tools can foster innovation and problem-solving across various domains, as it lowers the barrier to entry for individuals seeking to harness the power of ML without deep technical expertise.
* The library addresses the AutoML problem as a whole, considering the interconnections between sub-problems rather than treating each task in isolation. This holistic perspective is valuable because the performance of an AutoML system depends on the interplay between its various components, such as data pre-processing, feature engineering, model selection, hyperparameter optimization, and neural architecture search.
* The paper presents a thorough evaluation of HoNCAML, including both qualitative and quantitative assessments. The qualitative comparison with other AutoML libraries across various dimensions provides a clear understanding of HoNCAML's capabilities and limitations. The quantitative benchmark, conducted on a diverse set of public datasets using rigorous methodology, demonstrates the library's competitive performance and stability.
* HoNCAML's modular and extensible design, along with its open-source nature, facilitates future improvements and customization by the community. This openness encourages collaboration and allows researchers and practitioners to adapt the library to their specific needs, potentially accelerating the development of novel AutoML techniques and applications.

Weaknesses:
* HoNCAML currently only accepts tabular data as input, which may limit its applicability to certain domains that rely heavily on other data types, such as images, text, or time series. Extending the library's capabilities to handle a wider range of data types would increase its versatility and potential impact.
* The paper mentions that HoNCAML does not yet include a feature engineering module, which is a crucial step in many ML pipelines. The absence of this functionality may limit the library's performance on datasets that require significant feature transformations or the creation of new, informative features.

**Potential Impact On The Field Of Automl:**

The introduction of HoNCAML could have a notable impact on the field of AutoML by making machine learning more accessible to a wider range of users, including those with limited coding skills. The library's no-code approach and versatile interface options (CLI and GUI) lower the barrier to entry for individuals seeking to harness the power of ML without deep technical expertise.

**Review Confidence:**

3

**Review Rating:**

7

**Review Summary:**

In summary, HoNCAML is a promising open-source AutoML library that aims to democratize machine learning through its user-friendly, no-code approach and holistic consideration of AutoML sub-problems. The paper's comprehensive evaluation demonstrates the library's competitive performance and stability. Despite some limitations, such as the current restriction to tabular data and the lack of a developer's API and feature engineering module, HoNCAML's accessibility, extensible design, and open-source nature make it a valuable contribution to the field.

**Technical Quality And Correctness:**

The paper presents HoNCAML as a well-designed and thoroughly evaluated AutoML library. The authors have taken a holistic approach to address the AutoML problem, considering the interconnections between sub-problems. The library's modular and extensible design, along with its open-source nature, demonstrates a commitment to technical quality and facilitates future improvements and customization by the community.

---

### Official Review · Reviewer_Gbbw · 2024-04-03

**Potential Impact On The Field Of Automl Rating:** 3
**Technical Quality And Correctness Rating:** 2
**Clarity Rating:** 3
**Actions Required To Increase Overall Recommendation:** 1. More random seeds for all results …

**Summary Of Contributions:**

The paper introduces HoNCAML, a new AutoML tool that brings together many benefits of previous AutoML tools, incl. different interfaces (no-code GUI, CLI and code), a large set of model classes (incl. traditional ML) and optimizers (BO(HB) and CMAES). The tool is open-source and available under a BSD license. In the experiments, the authors compare HoNCAML against other AutoML tools on different tabular classification and regression datasets.

**Clarity:**

Overall, the paper is well written and has only minor clarity problems:

1. Section 1, please cite the AutoML tools instead of providing links to their websites
1. The pitch of HoNCAML is a versatile usage. However, if I understand it correctly, it can only be applied to tabular benchmarks. However, this is a major limiting factor. This needs to be clarified in the paper.
1. The description of algorithm selection is misleading. According to Rice 1976, it is a meta-learning problem to predict the best algorithm given a task. I believe that this should clarified in 3.3.
1. The optimizers are only imported from RayTune and not self-implemented. This should be clarified.
1. The authors mention NAS, but do not mention anything related to gradient-based approaches such as DARTS.
1. Section 4.1 has a sentence that is repeated (more or less) twice.
1. The best-performing results in Table 2 and Table 3 could highlighted (e.g., bold-face) to increase readability.
1. I do not understand how the statistical tests were applied. If the p-value is smaller than 0,05, I would conclude that there is a statistical difference and thus, HoNCAML could be statistically worse than one of the other AutoML tools. Please clarify.

**Overall Review:**

Also there might be a reasonable impact of HoNCAML, the paper lacks clarity and empirical soundness (most importantly: more random seeds). On only one of the datasets, HoNCAML turns out to be the winner and in this case (Sensory), nearly all tools achieved the same performance.

**Potential Impact On The Field Of Automl:**

Overall, HoNCAML is a step forward to a fully-fledged and complete AutoML tool that addresses different users by providing different interfaces and search spaces (incl. NAS). From my point of view, this is the strongest point of HoNCAML. As discussed below, unfortunately, the empirical results are not very strong, and thus, it cannot be considered state-of-the-art. Therefore, it most likely will be cited for the way it is used and less for empirical performance.

**Review Confidence:**

5

**Review Rating:**

7

**Review Summary:**

Reading the paper, I would agree that the HoNCAML fulfills the following AUTOML24 criteria (according the CfP) to a certain degree:

* It is a novel system that has features or application domains that were not available beforehand.
* It already has an established user base (shown by stars on github, active commit history by several developers, an active issue tracker, etc.)
* It is an open-source software package with an open-source software license that allows users to easily use and contribute to it.

However, it fails to deliver:
* It achieves excellent performance on the addressed application domains.

**Technical Quality And Correctness:**

Overall, the tool is well described, but there are various comments and questions I would like to raise:

1. What is meant by "correlation between its sub-problems"? Data can be correlated, but how are problem correlated?
1. The authors claim that "The library offers a wide range of state-of-the-art methods to solve this challenge". However, I wonder how HoNCAML decides which one to use in practice. Obviously, HoNCAML cannot leave this decision to its users.
1. The datasets (as listed in Table 1b) are fairly simple datasets. It is not very convincing that the authors selected datasets according which datasets can be read by HoNCAML.
1. All results are based on a single random seed (42). However, all these tools are highly randomized. Therefore, robust results can only be expected by running them with several random seeds -- typically, one would use 10 random seeds.
1. The 180sec timeout is fairly small.
1. Related to the previous three points, I would strongly recommend using the AMLB setting and setup for comparing AutoML tools. This provides a well-established set of datasets, sufficient repeated measurements, and reasonable timeouts. https://arxiv.org/abs/2207.12560
1. Although the authors emphasize over and over again that HoNCAML provides so many models, the results in Table 2 and 3 do not support that HoNCAML really benefits from it.

---

### Official Review · Reviewer_j2qD · 2024-04-04

**Potential Impact On The Field Of Automl Rating:** 3
**Technical Quality And Correctness Rating:** 2
**Clarity Rating:** 3

**Summary Of Contributions:**

The paper presents a publicly available software solution, HoNCAML, that enables "no-code" AutoML. It explains the motivations and design principles used to build the solution. HoNCAML focuses on three tasks: hyper-parameter optimization, algorithm selection, and neural architecture search. The paper includes evaluations of the solution, comparisons with other tools, and instructions on how to get started with HoNCAML.

**Details Of Ethical Concerns:**

The authors could use the appendix to explain how this tool can or will mitigate discrimination, bias, and fairness. For instance, in addition to the raw results, the user could be provided with additional information to better understand the system's results. Other concerns include privacy, security, safety, legal compliance, and the prevention of potentially harmful insights, methodologies, or applications.

**Actions Required To Increase Overall Recommendation:**

- Improve the qualitative and quantitative evaluation of the solution.
- What are the innovations compared to other existing solutions?
- Discuss how the solution can be integrated with other solutions.
- Address other comments in the boxes above.

**Clarity:**

The paper is well-written. Its outline and flow can be improved to better emphasize the important aspects of the solution.

**Ethics And Accessibility Rating:**

["Yes, regarding discrimination / bias / fairness", "Yes, regarding privacy / security / safety", "Yes, regarding legal compliance (e.g., GDPR, copyright, terms of use)", "Yes, regarding potentially harmful insights, methodologies, or applications"]

**Overall Review:**

Strengths: The AutoML research community and the rest of society can benefit from another solution for "no-code." The current version of the solution is compelling and could be developed into a robust solution.

Weaknesses: The paper must make a stronger claim or present key differentiators for choosing this solution over existing tools. What innovations this solution has needs to be clarified. The qualitative and quantitative evaluations of the solution could be more robust. The authors should also investigate and cite existing literature on designing similar software solutions.

**Potential Impact On The Field Of Automl:**

The paper is important for the AutoML community and could be used as a source for comparing "no-code" AutoML solutions.

**Review Confidence:**

5

**Review Rating:**

4

**Review Summary:**

The paper is well-written, and the proposed solution is compelling. However, the description of the solution, motivations, and potential novelty should be improved. For instance, if one of the highlights is the GUI, then the authors should conduct studies to validate and support the design decisions. How does it enable the "no-code" paradigm? For instance, what other user studies on similar software systems have concluded about the best practices for developing this kind of solution?

The current version of the paper can be significantly improved to result in a solid paper in the future.

**Technical Quality And Correctness:**

The solution's description is sound. However, the evaluation, the related work section, and its comparison with other existing tools must be revised. After reading the paper, it needs to be clarified why one would choose the proposed solution over other available tools and services. More technical details, e.g., its application of NAS, are needed. The authors should also highlight any proposed innovation and support their design decisions by referring to existing principles and studies for software design.

---

### Meta-Review · Area_Chair_H2bU · 2024-04-22

**Paper Recommendation:** Accept
**Confidence:** 4

**Metareview:**

This work proposes a new AutoML library HoNCAML that comes with GUI and CLI interface, to address user cases with lower coding efficiency. It aims to address all the important steps for building a machine learning pipeline: data preprocessing, feature engineering (future work), model selection (including NAS) and hyperparameter optimization. The empirical results on 12 tabular datasets shows competitive performance compared with other AutoML libraries.

The strengths of the paper are the interfacing and reproducibility. It pushes the publicly available AutoML library one step further with its GUI and CLI Interfaces for the Low Code/No Code user cases (commercial Low Code/No Code tools do exist). It has very good reproducibility as pointed out by our reproducibility reviewer.

The weaknesses are the relatively limited evaluations and feature incompleteness. The current evaluation is based on 12 tabular datasets. Even though the library supports NAS, there is no comparison to the previous NAS methods on NAS problems. Feature engineering is missing at the time of submission. These points are slightly mitigated by the active development of integration of AMLB (visible at Github) and feature engineering (not publicly visible).

I am inclined  to accept this paper for its unique position on the interfacing, good reproducibility and active development which mitigates the weakness to some degree.

---

### Decision · Program_Chairs · 2024-04-29

**Decision:**

Accept

**Comment:**

Thank you for submitting your paper. We are happy to tell you that we accept your paper to the main track. See you in Paris.